# Study on the Mathematical Model of Vacuum Breaker Valve for Large Air Mass Conditions

**Xiao-ying Zhang [1], Cheng-yu Fan [2], Xiao-dong Yu [2],*, Jian Zhang [1,2], Jia-wen Lv [2] and Ting-yu Xu [2]**

[1]   College of Hydraulic and Civil Engineering, Xinjiang Agricultural University, Urumqi 830052, China
[2]   College of Water Conservancy and Hydropower Engineering, Hohai University, Nanjing 210098, China
*   Correspondence: yuxiaodong_851@hhu.edu.cn

**Abstract:** The mathematical model of vacuum breaker valve is significant to the protection scheme. The more accurate the vacuum breaker valve model, the more reliable the calculation results. In this study, the application conditions of the air valve model are analyzed according to the assumptions used in the derivation, and the contradictions between these assumptions are proposed. Then, according to the different working characteristics between the vacuum breaker valve on the siphon outlet pipe and the air valve, the vacuum breaker valve model is deduced based on the modified assumptions. In the derivation process, the thermodynamic change of the gas in the vacuum breaker valve is assumed to follow the isentropic process rather than an isothermal process, and the water level in the vacuum breaker valve is considered to be changeable. An engineering case is introduced, and the results calculated according to the vacuum breaker valve model are compared with those resulting from the air valve model. The results indicate that the vacuum breaker valve model is suitable for large air mass conditions and can provide a theoretical basis for the numerical simulation and settings of vacuum breaker valves.

**Keywords:** vacuum breaker valve; air valve; mathematical model; large air mass condition

## 1. Introduction

The head loss of the siphon pipe is small, the flow control in the siphon pipe is convenient when the pumps are powered off, and the water level at the end of the siphon pipe can be lowered [1–3]. Therefore, many large pump systems with low heads and large flow usually have siphon outlet pipes behind the pumps. To prevent serious negative pressure damage when the pumping station is powered off, a vacuum breaker valve is usually installed at the top of the siphon pipe for security protection. As a kind of safety device, the vacuum breaker valve can be both electronically and automatically controlled. If it is in electronic control mode, it will open immediately as soon as the power failure happens, while if it is in automatic control mode, it will work as long as the piezometric head in the pipe at the location of the vacuum breaker valve decreases sharply and becomes smaller than the set intake pressure [4]. The air sucked into the vacuum breaker valve releases the vacuum, preventing the pipe from liquid column separation and rejoining.

In recent years, many studies have proposed water hammer protection with vacuum breaker valves in water supply projects [5–7]. Lee and Leow proposed an improved numerical model and the calculation method of vacuum breaker valve, using the calculation of the pressure pulsation of gas-containing fluid in pumping station systems [8]. Numerical experiments showed that air valves installed at the top of water pipelines under mass flow could be used to reduce the magnitude of the negative pressure [9–13]. Lingireddy et al. found that if the air was released too quickly, the final air release through the air vacuum valves would produce a pressure surge. Therefore, the air release vacuum must be properly designed to avoid excessive pressure surge [14]. Li et al. characterized the

dynamics of the vacuum breaker valves and other kinds of valves, the simulation results showed that the large pressure spikes can be generated in the vacuum breaker valve [15]. Ramezani and Karney used the basic water hammer theory to semi-analytically explore the effects of friction to the vacuum breaker valves [16]. Although the research on the influence of vacuum breaker valves on the secondary transient events were conducted in the experimental studies [5,7,14,15], the rules of the physical phenomena had not been clearly explained. Cabrera et al. explained how a specific position of the air vacuum valve affects the transient response of the system [17]. Since the installation position and air intake mode of the vacuum breaker valve and the air valve were similar, the mathematical model of the vacuum breaker valve was still calculated with the air valve model in most projects. Therefore, the research on air valves can provide reference for the development of a vacuum breaker valve [18–20]. Jönsson raised recommendations for the correct installation position of air valves to reaerate the pipeline [21]. Zhou et al. investigated various situations of pressure changes occurring in pipelines with several air pockets without air release valves installed at local high points [22–24]. Considering air valve characteristics, riser dimensions, driving head and allowable working pressure, methodologies for determining safe filling rates for the pipeline were proposed by Phu D. Tran [25]. Albertson et al. investigated pressure transients during filling for the cases of air vented through an orifice plate or a large-orifice air valve [26,27]. The situation of filling a pipeline with an orifice plate or a small-orifice air valve was investigated by several researchers [28–30].

Vacuum breaker valves and air valves are both negative pressure protection devices, whereas they still have some different features. Firstly, the intake pressures of these two valves are different. The air inlet pressure of the air valve is 0 m, while the air inlet pressure of the vacuum breaker valve can be set according to actual engineering needs. Secondly, the valve sizes are different. More air valves are needed to meet the requirements of water hammer protection due to the diameter of each single air valve and the amount of the air passing through each valve are both small. In addition, the amount of the air passing through each valve is usually small. It is difficult for the air valve discharging out the air that sucked into the pipe as a result of small valve size. So, the air valve is usually used as an auxiliary water hammer protection device for long-distance pipes. On the contrary, the size of a vacuum breaker valve can be much larger, so it does not need to install too many vacuum breaker valves in a project. In most cases there is still less air passing through the vacuum breaker valve [20,31,32], it is reasonable to use the air valve model for one-dimensional numerical simulation of vacuum breaker valve. However, as to the large air mass conditions (for example, only one vacuum breaker valve is set at the top of each siphon pipe, which is of short distance, large flow, and low water level of the outlet sump), the amount of the air passing through each valve can be much larger. As a result, the assumptions of the air valve model cannot be satisfied. The air valve model is not suitable for the simulation of the vacuum breaker valve anymore. Considering the importance of the accuracy in the numerical model, it directly relates to the reliability of the simulation results for a protection scheme. It is essential to deduce the vacuum breaker valve model, which is suitable for large air mass conditions. In this study, the thermodynamic change of the air in the vacuum breaker valve is assumed to follow the isentropic process, and the water level of the vacuum breaker valve is considered to be variable. According to these modified assumptions, the vacuum breaker valve model is derived based on the air valve model. The vacuum breaker valve model can provide a theoretical basis for the water hammer protection scheme in a large water supply project.

## 2. Methods

### 2.1. Problems with the Air Valve Mathematical Model

The schematic diagram of the air valve is shown in Figure 1. The air valve model can be deduced based on the following four assumptions:

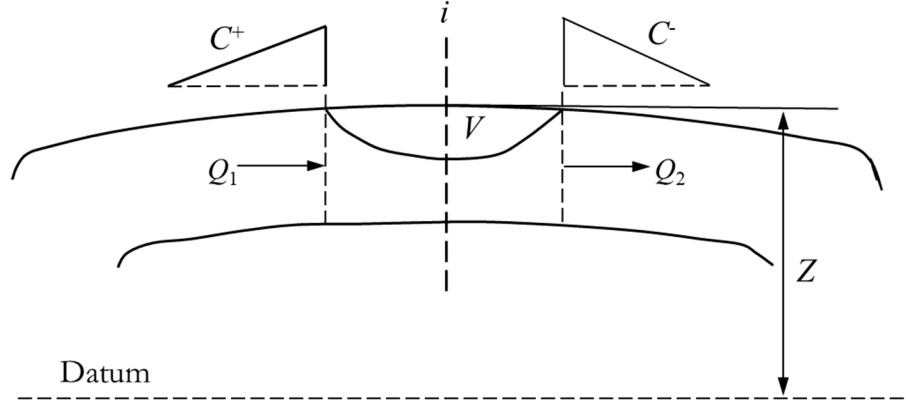

**Figure 1.** Schematic diagram of an air valve.

Assumption 1: The length of the valve hole is so short, so the thermodynamic change in the air going in or out of the air valve can be assumed to follow the isentropic process.

Assumption 2 and Assumption 3: The amount of the air sucked into the air valve is small, so the thermodynamic change of the air in the air valve can be assumed to follow the isothermal process, remaining at the atmospheric temperature. The water level in the air valve can be considered to be unchangeable and constant with the height of the pipe top at the location of the air valve.

Assumption 4: The location of the air in the air valve is assumed to be fixed at the top of the air valve, so the air valve can be considered as a node.

(1) According to Assumption 1 [8,32]:

If $0.5283p_a < p < p_a$, then the air will be sucked in with subsonic speed:

$$\dot{m} = C_{in}A_{in} \sqrt{7p_a\rho_a\left(\left(\frac{p}{p_a}\right)^{1.4286} - \left(\frac{p}{p_a}\right)^{1.7143}\right)} \tag{1}$$

in which $\dot{m}$ is the mass flow of the air passing through the air valve, kg/s; $C_{in}$ is the inflow coefficient of the gas; $A_{in}$ is the cross-sectional area of the valve hole in the inflow direction, m²; $p_a$ is the atmospheric pressure, Pa; $\rho_a$ is the atmospheric density, kg/m³; and $p$ is the pressure of the gas in the air valve, Pa.

If $p \leq 0.5283p_a$, the air will be sucked in with critical velocity:

$$\dot{m} = C_{in}A_{in}\frac{0.6847p_a}{\sqrt{RT_a}} \tag{2}$$

in which $\dot{m}$ is the mass flow of the air passing through the air valve, kg/s; $C_{in}$ is the inflow coefficient of the gas; $p_a$ is the atmospheric pressure, Pa; $R$ is the gas constant, J/(kg·K), and $T_a$ is the atmospheric temperature, K.

If $p_a < p < \frac{p_a}{0.5283}$, the air will be discharged out with subsonic speed:

$$\dot{m} = -C_{out}A_{out}p \sqrt{\frac{7}{RT}\left(\left(\frac{p_a}{p}\right)^{1.4286} - \left(\frac{p_a}{p}\right)^{1.7143}\right)} \tag{3}$$

in which $\dot{m}$ is the mass flow of the air passing through the air valve, kg/s; $C_{out}$ is the outflow coefficient of the gas; $A_{out}$ is the cross-sectional area of the valve hole in the outflow direction, m²; $p$ is the pressure of the gas in the air valve, Pa; Pa and $T$ is the temperature of the gas in the air valve, K.

If $p \geq \frac{p_a}{0.5283}$, the air will be discharged out with critical velocity:

$$\dot{m} = -C_{out}A_{out}p\frac{0.6847}{\sqrt{RT}} \tag{4}$$

in which $\dot{m}$ is the mass flow of the air passing through the air valve, kg/s; $C_{out}$ is the outflow coefficient of the gas; $A_{out}$ is the cross-sectional area of the valve hole in the outflow direction, $m^2$; $p$ is the pressure of the gas in the air valve, Pa; $R$ is the gas constant, $J/(kg \cdot K)$, and $T$ is the temperature, K.

(2) According to Assumption 3 and Assumption 4, the compatibility equation of the MOC (Method of Characteristics), the continuity equation, the pressure equation and the equation of state for an ideal gas [8]:

$$p\left\{V_0 + 0.5\Delta t\left[Q_{20} - Q_{10} - \left(\frac{C_P}{B_P} + \frac{C_M}{B_M}\right) + \left(\frac{1}{B_P} + \frac{1}{B_M}\right)\left(\frac{p-p_a}{\gamma} + Z\right)\right]\right\} = \left[m_0 + 0.5\Delta t\left(\dot{m}_0 + \dot{m}\right)\right]RT \tag{5}$$

in which $p$ is the pressure of the gas in the air valve, Pa; $V_0$ is the volume of the gas at the beginning of the time step, $m^3$; $\Delta t$ is the time step, s; $Q_{10}$ is the inflow at the beginning of the time step, $m^3/s$; $Q_{20}$ is the outflow at the beginning of the time step, $m^3/s$; the coefficients $C_P$, $B_P$, $C_M$, and $B_M$ = constants; $\gamma$ is the specific gravity of water, $kg/(m^2 \cdot s^2)$; $Z$ is constant to be the height of the pipe top at the location of the air valve, m; $m_0$ is the mass of the gas in the air valve at the beginning of the time step, kg; $\dot{m}$ is the mass flow of the air passing through the air valve, kg/s; and $\dot{m}_0$ is the mass flow of the air passing through the air valve at the beginning of the time step, kg/s.

(3) According to Assumption 2, the temperature of the gas in the air valve $T$ in Equations (3)–(5) is equal to the atmospheric temperature $T_a$. $p$ and $\dot{m}$ are the only two unknown parameters in Equation (5), and $\dot{m}$ can be calculated according to Equations (1)–(4), so $p$ can be solved.

As shown in Figure 2, at the beginning of the time step, the mass of the air in the air valve is $m_1$ with the state of $p_1$, $V_1$ and $T_1$, while at the end of the time step, the mass of the air in the air valve changes to $m_1 + \Delta m = m_2$ with the state of $p_2$, $V_2$ and $T_2$.

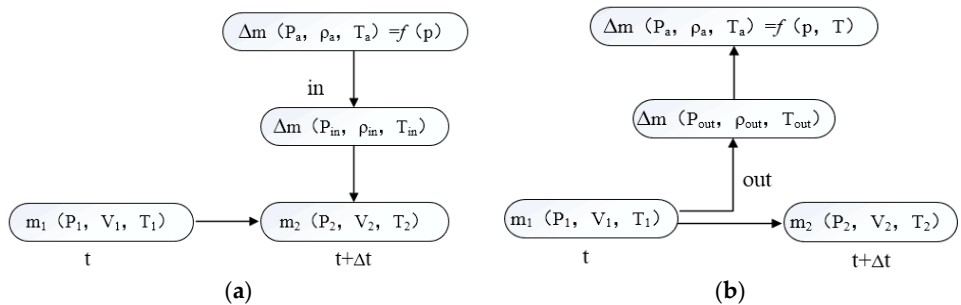

**Figure 2.** Processes of air passing through the air valve: (**a**) Gas sucking in; (**b**) gas discharging out.

According to Assumption 1, the thermodynamic change in the $\Delta m$ air going in or out of the air valve follows the isentropic process. $p_a > p_{in}$ should be satisfied if the air can be sucked in the air valve, according to the polytropic equation, $T_a > T_{in}$, while $p_a < p_{out}$ should be satisfied if the air can be discharged out of the air valve, $T_a < T_{out}$. That is, the temperature of the air sucked in or discharged out of the air valve should be different from the atmospheric temperature. However, according to Assumption 2, as the thermodynamic change in the gas in the air valve follows the isothermal process, remaining as the atmospheric temperature, $T_1 = T_2 = T_a$. So $T_{in} \neq T_2$ and $T_{out} \neq T_2$, which are exact contradictions.

In summary, there are contradictions between Assumption 1 and Assumption 2 of the air valve model. Under these assumptions, the state of the air sucked in or discharged out of the air valve is different from that of the air in the air valve. The air pressure $p$ in Equations (1)–(4) should be different from that in Equation (5). Actually, the state of the gas after isentropic inhalation or before discharge cannot be obtained by the isothermal change in the gas in the air valve; the gas requires a process transition of thermodynamic changes. Therefore, it is assumed that the intake and exhaust processes of the air cannot be completely simulated under Assumptions 1 and 2. The simultaneous solutions of these equations are not reasonable.

For the vacuum break valve arranged on the siphon outlet pipe, since the intake air amount is relatively larger, it is more difficult to achieve sufficient heat exchange between the gas and water, and Assumption 2 of the air valve is more difficult to achieve. When a large amount of air is taken into the pipe, the air valve model cannot be applicable to simulate the vacuum breaker valve. It is necessary to reintroduce the mathematical model of the vacuum break valve to accommodate the large air mass conditions of the outlet pipe.

*2.2. Mathematical Model of the Vacuum Breaker Valve*

According to the air valve model assumptions and discussion in the previous section, the air valve model is only suitable for small air mass conditions, and there are contradictions between Assumption 1 and Assumption 2. To solve the contradictions and make the model fit large air mass conditions, the vacuum breaker valve model is deduced based on the air valve model according to the following four modified assumptions:

Assumption 1: The length of the valve hole is so short, so the thermodynamic change in the air going in or out of the vacuum breaker valve can be assumed to follow the isentropic process.

Assumption 2 and Assumption 3: The amount of the air sucked into the vacuum breaker valve is large, so the thermodynamic change in the gas in the vacuum breaker valve can be assumed to follow the isentropic process, and the water level in the vacuum breaker valve can be considered to be variable.

Assumption 4: The location of the gas in the vacuum breaker valve is assumed to be fixed at the top of the vacuum breaker valve, so the vacuum breaker valve can be considered as a node.

(1) According to Assumption 1:

Assumption 1 is the same as that of the air valve model, so Equations (1)–(4) can be applied for the vacuum breaker valve model.

(2) The thermodynamic change in the gas in the vacuum breaker valve is assumed to follow an isentropic process rather than an isothermal process. According to Assumption 1 and Assumption 2:

For the initial steady state, the vacuum breaker valve is closed, and there is no air in the pipe.

For the end of the first-time step, the mass of the air in the vacuum breaker valve changes to be $\Delta m_1$. According to the polytropic equation and the equation of state for an ideal gas:

$$p_a V_{1,a}{}^{1.4} = p_{1,\text{in}} V_{1,\text{in}}{}^{1.4} = p_{1,2} V_{1,2}{}^{1.4} \tag{6}$$

$$\frac{p_a V_{1,a}}{T_a} = \frac{p_{1,\text{in}} V_{1,\text{in}}}{T_{1,\text{in}}} = \frac{p_{1,2} V_{1,2}}{T_{1,2}} = \Delta m_1 R \tag{7}$$

For the parameters with two subscripts, the first subscript represents the number of the time step, while the second subscript has the same meaning as before. According to Equation (6) and (7), obviously if $p_{1,\text{in}} = p_{1,2}$, $V_{1,\text{in}} = V_{1,2}$, and $T_{1,\text{in}} = T_{1,2}$.

For the end of the second time step, on the one hand, the volume of the $\Delta m_1$ gas in the vacuum breaker valve changes to be $V_{2,y}$. Then:

$$p_{1,2} V_{1,2}{}^{1.4} = p_{2,2} V_{2,y}{}^{1.4} \tag{8}$$

$$\frac{p_{1,2} V_{1,2}}{T_{1,2}} = \frac{p_{2,2} V_{2,y}}{T_{2,2}} = \Delta m_1 R \tag{9}$$

By substituting Equation (6) and Equation (7) into Equation (8) and Equation (9):

$$T_{2,2} = \left(\frac{p_{2,2}}{p_a}\right)^{1-1/1.4} T_a \tag{10}$$

On the other hand, for the $\Delta m_2$ gas sucked into the pipe during $\Delta t \sim 2\Delta t$:

$$p_a V_{2,a}^{1.4} = p_{2,\text{in}} V_{2,\text{in}}^{1.4} \tag{11}$$

$$\frac{p_a V_{2,a}}{T_a} = \frac{p_{2,\text{in}} V_{2,\text{in}}}{T_{2,\text{in}}} = \Delta m_2 R \tag{12}$$

By substituting Equation (11) into Equation (12):

$$T_{2,\text{in}} = T_a \left( \frac{p_{2,\text{in}}}{p_a} \right)^{1-1/1.4} \tag{13}$$

By comparing Equation (10) and Equation (13), if $p_{2,\text{in}} = p_{2,2}$, then $T_{2,\text{in}} = T_{2,2}$.

The same conclusions can be found if there is $\Delta m_2$ gas discharged out of the pipe during $\Delta t \sim 2\Delta t$. By substituting the subscript "in" in Equations (11)–(13) with "out": If $p_{2,\text{out}} = p_{2,2}$, then $T_{2,\text{out}} = T_{2,2}$.

In addition, as $p_a V_{1,a}^{1.4} = p_{2,2} V_{2,y}^{1.4}$ according to Equation (6) and Equation (8) and $p_a V_{2,a}^{1.4} = p_{2,\text{in}} V_{2,\text{in}}^{1.4}$ according to Equation (11), if $p_{2,\text{in}} = p_{2,2}$:

$$\frac{p_a}{p_{2,2}} = \left( \frac{V_{2,y}}{V_{1,a}} \right)^{1.4} = \left( \frac{V_{2,\text{in}}}{V_{2,a}} \right)^{1.4} = \left( \frac{V_{2,y} + V_{2,\text{in}}}{V_{1,a} + V_{2,a}} \right)^{1.4} = \left( \frac{V_{2,2}}{V_{1,a} + V_{2,a}} \right)^{1.4} \tag{14}$$

However, as $\frac{p_a V_{1,a}}{T_a} = \frac{p_{2,2} V_{2,y}}{T_{2,2}} = \Delta m_1 R$ according to Equation (7) and Equation (9) and $\frac{p_a V_{2,a}}{T_a} = \frac{p_{2,\text{in}} V_{2,\text{in}}}{T_{2,\text{in}}} = \Delta m_2 R$ according to Equation (12), if $p_{2,\text{in}} = p_{2,2}$ and $T_{2,\text{in}} = T_{2,2}$:

$$\frac{p_a(V_{1,a} + V_{2,a})}{T_a} = \frac{p_{2,2}\left( V_{2,y} + V_{2,\text{in}} \right)}{T_{2,2}} = \frac{p_{2,2} V_{2,2}}{T_{2,2}} = (\Delta m_1 + \Delta m_2)R \tag{15}$$

For the end of the *n*-th time step, according to Equation (14) and (15), the state of the gas in the vacuum breaker valve at the end of the second time step is the same as that of the $\Delta m_1 + \Delta m_2$ gas directly sucked into the vacuum breaker valve following the isentropic process. Therefore, the gas in the vacuum breaker valve at the end of the *n*-th time step can be assumed as follows:

$$\frac{p_a}{p_{n,2}} = \left[ \frac{V_{n,2}}{\left( \sum\limits_{x=1}^{n} V_{x,a} \right)} \right]^{1.4} \tag{16}$$

$$\frac{p_a \sum\limits_{x=1}^{n} V_{x,a}}{T_a} = \frac{p_{n,2} V_{n,2}}{T_{n,2}} = \sum\limits_{x=1}^{n} \Delta m_x R \tag{17}$$

For the end of the *n* + 1-th time step, on the one hand, for the $\sum\limits_{x=1}^{n} \Delta m_x$ gas in the vacuum breaker valve at the end of the *n*-th time step:

$$T_{n+1,2} = \left( \frac{p_{n+1,2}}{p_a} \right)^{1-1/1.4} T_a \tag{18}$$

On the other hand, for the $\Delta m_{n+1}$ gas sucked into the pipe during $n\Delta t \sim (n+1)\Delta t$:

$$T_{n+1,\text{in}} = T_a \left( \frac{p_{n+1,\text{in}}}{p_a} \right)^{1-1/1.4} \tag{19}$$

By comparing Equation (18) and Equation (19), if $p_{n+1,in} = p_{n+1,2}$, then $T_{n+1,in} = T_{n+1,2}$.

There is no contradiction between the modified Assumption 1 and Assumption 2. The state of the gas in the vacuum breaker valve at any time is the same as that of the gas with the same mass directly sucked into the vacuum breaker valve following the isentropic process. According to Equation (18), the gas temperature in the vacuum breaker valve is as follows:

$$T = T_a \left( \frac{p}{p_a} \right)^{1-1/1.4} \tag{20}$$

(3) According to Assumption 3 and Assumption 4:

The schematic diagram of the vacuum breaker valve is shown in Figure 3. The compatibility equation of the MOC, the continuity equation, the pressure equation at the location of the vacuum breaker valve and the equation of water level change separately are as follows:

$$\begin{aligned} C^+ : H &= C_P - B_P Q_1 \\ C^- : H &= C_M + B_M Q_2 \end{aligned} \tag{21}$$

$$\frac{dV}{dt} = Q_2 - Q_1 \tag{22}$$

$$\frac{p - p_a}{\gamma} = H - Z \tag{23}$$

$$dV = -A dZ \tag{24}$$

in which $H$ is the piezometric head at the location of the vacuum breaker valve, m, $C_P$, $B_P$, $C_M$, $B_M$ are constants; $Q_1$ is the inflow of the section; $Q_2$ is the discharge flow of the section; $V$ is the volume of gas, $\gamma$ is the specific gravity of water, $kg/(m^2 \cdot s^2)$; $p$ is the pressure of the gas in the air valve, Pa; $p_a$ is the atmospheric pressure, Pa; $Z$ is constant to be the height of the pipe top at the location of the air valve, m; and $A$ is the cross-sectional area of the valve hole, $m^2$.

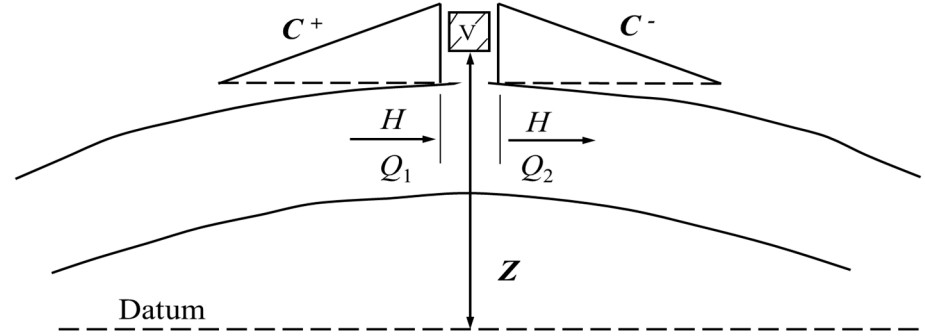

**Figure 3.** Schematic diagram of the vacuum breaker valve.

By substituting Equation (21) into Equation (23):

$$\frac{p - p_a}{\gamma} = C_P - B_P Q_1 - Z = C_M + B_M Q_2 - Z \tag{25}$$

According to Equation (22) and Equation (24):

$$Z = -\frac{0.5 \Delta t (Q_2 + Q_{20} - Q_1 - Q_{10})}{A} + Z_0 \tag{26}$$

in which $Z_0$ is the water level in the vacuum breaker valve at the beginning of the time step, m.

According to Equation (25) and Equation (26):

$$(Q_2 - Q_1)\left(B_P B_M + \frac{0.5\Delta t(B_P + B_M)}{A}\right)$$
$$= (B_P + B_M)\left(\frac{p - p_a}{\gamma} + Z_0 - \frac{0.5\Delta t(Q_{20} - Q_{10})}{A}\right) - (B_P C_M + B_M C_P) \tag{27}$$

By substituting Equation (22) and Equation (27) into the equation of state for an ideal gas:

$$p\left\{V_0 + 0.5\Delta t\left[Q_{20} - Q_{10} + \frac{(B_P + B_M)\left(\frac{p - p_a}{\gamma} + Z_0 - \frac{0.5\Delta t(Q_{20} - Q_{10})}{A}\right) - (B_P C_M + B_M C_P)}{\left(B_P B_M + \frac{0.5\Delta t(B_P + B_M)}{A}\right)}\right]\right\}$$
$$= \left[m_0 + 0.5\Delta t(\dot{m}_0 + \dot{m})\right]RT \tag{28}$$

in which $p$ is the pressure of the gas in the air valve, Pa; $V_0$ is the volume of the gas at the beginning of the time step, m$^3$; $\Delta t$ is the time step, s; $Q_{10}$ is the inflow at the beginning of the time step, m$^3$/s; $Q_{20}$ is the outflow at the beginning of the time step, m$^3$/s; the coefficients $C_P$, $B_P$, $C_M$, and $B_M$ = constants; $\gamma$ is the specific gravity of water, kg/(m$^2 \cdot$ s$^2$); $p$ is the pressure of the gas in the air valve, Pa; $p_a$ is the atmospheric pressure, Pa; $A$ is the cross-sectional area of the valve hole, m$^2$; $Z_0$ is the water level in the vacuum breaker valve at the beginning of the time step, m; $Z$ is constant to be the height of the pipe top at the location of the air valve, m; $m_0$ is the mass of the gas in the air valve at the beginning of the time step, kg; $\dot{m}$ is the mass flow of the air passing through the air valve, kg/s; and $\dot{m}_0$ is the mass flow of the air passing through the air valve at the beginning of the time step, kg/s.

For the temperature of the gas in the vacuum breaker valve $T$ in Equation (3), Equation (4) and Equation (28) can be calculated according to Equation (20); furthermore, $p$ and $\dot{m}$ are the only two unknown parameters in Equation (28). $\dot{m}$ can be calculated according to Equations (1)–(4), and $p$ can be calculated.

By comparing Equation (5) and Equation (28), if the air mass sucked into the pipe is small, $Z \to Z_0$ and $A \to \infty$ should be satisfied in Equation (28). Then, Equation (28) can be simplified with the same form as Equation (5). That is, the air valve model can be regarded as a special case of the vacuum breaker valve model.

## 3. Results

### 3.1. Project Description

As shown in Figure 4, the water supply system is mainly composed of the following components: Suction sump, outlet sump, pump, siphon pipe and vacuum breaker valve. Meanwhile, the parameters of the water supply system are listed in Table 1. From the table, we can see that the water levels of the suction sump and outlet sump are 28 m and 38.5 m, respectively. The siphon has a horizontal length of 105 m and a diameter of 2.4 m. The vacuum break valve is placed at the top of the siphon, with a −3-m intake pressure and 0.3-m diameter. Separately, the vacuum breaker valve is numerically simulated according to the air valve model as well as the vacuum breaker valve model.

**Table 1.** System parameters.

| Water Level of Suction Sump (m) | 28.0 | Diameter of Vacuum Breaker Valve (m) | 0.3 |
|---|---|---|---|
| Water level of outlet sump (m) | 38.5 | Intake pressure of vacuum breaker valve (m) | −3.0 |
| Quantity of pipes | 1 | Quantity of pumps | 1 |
| Elevation of pipe center at outlet sump end (m) | 30.0 | Elevation of pump (m) | 25.0 |
| Elevation of pipe center at top of siphon pipe (m) | 37.0 | Rated head (m) | 12 |
| Horizontal length of pipe (m) | 105.0 | Rated flow (m$^3$/s) | 10 |
| Pipe diameter (m) | 2.4 | Rated rotational speed (r/min) | 250 |
| Design flow (m$^3$/s) | 10 | Rated motor power (kW) | 1600 |
| Elevation of vacuum breaker valve (m) | 40.0 | Flywheel moment (kg·m$^2$) | 3800 |

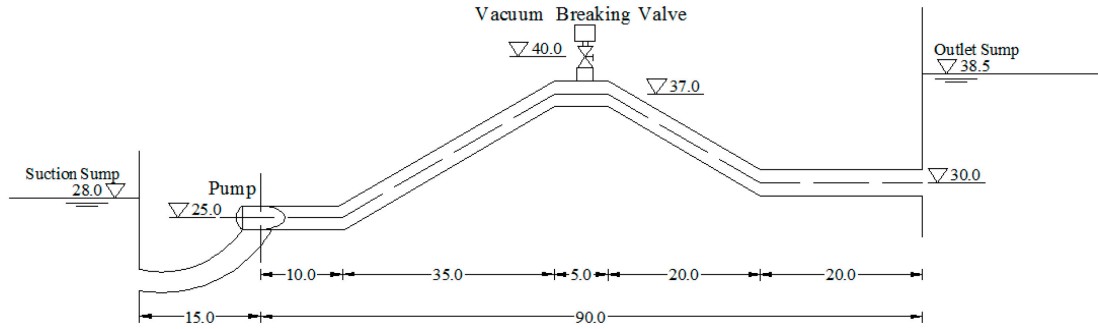

**Figure 4.** Water supply system layout.

### 3.2. Numerical Simulation of the Air Valve and Vacuum Breaker Valve

The mathematical model introduced in the previous sections was used to simulate the pumping system with a siphon outlet pipe. Based on the MOC, the computer models of the air valve and the vacuum breaker valve were encoded in the FORTRAN programming language, as shown in Figure 4. When simulating the vacuum breaker valve at the top of the siphon pipe with the air valve model and the vacuum breaker valve model, the calculation results are as shown in Figures 5–7.

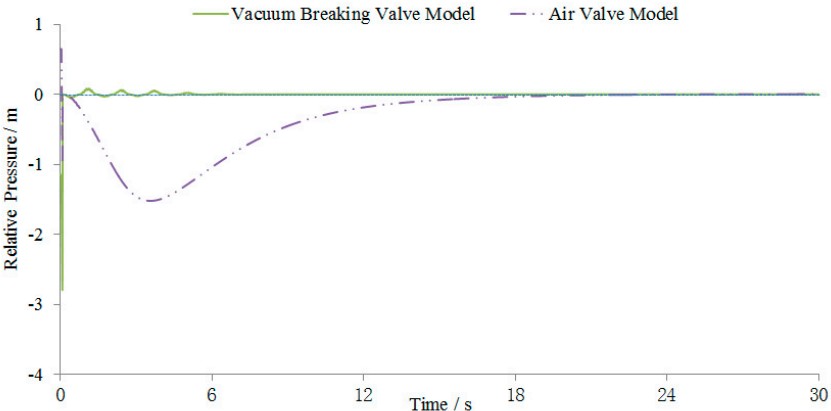

**Figure 5.** Relative pressure changes in the air in the valve.

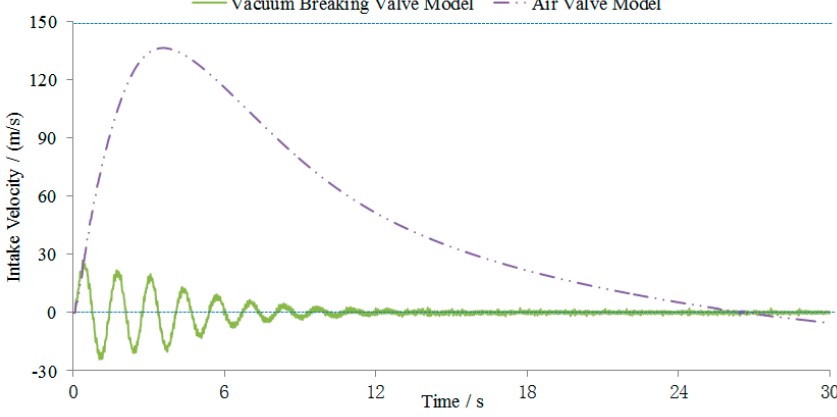

**Figure 6.** Intake velocity changes in the air through the valve.

The different relative pressure changes in the air in the pipe are shown in Figure 5. After a water pumping accident occurs, the vacuum at the top of the siphon pipe reaches the set limit, the vacuum breaker valve opens, and the gas pressure in the valve quickly rises to atmospheric pressure. When the air valve model is used to numerically simulate the vacuum breaker valve, the negative pressure wave is quickly transmitted to the vacuum breaker valve at the valve opening time, causing the relative

pressure of the gas to drop rapidly to −1 m at $t$ = 0.1 s, while at $t$ = 3.6 s, it is reduced to −2.5 m again, then slowly rises, the gas pressure tends to 0 m, and then remains unchanged. As the water level in the vacuum breaker valve is unchangeable according to the air valve model, the water level is higher than it should be, leading to a smaller relative pressure. Therefore, the relative pressure of the gas first decreases and then gradually rises to zero. When the vacuum breaker valve model is used for the simulation calculation, the relative pressure is suddenly reduced to −2.8 m at the valve opening time due to the negative pressure wave action, and then quickly rises to 0 m, and when $t \geq 5$ s, the gas pressure remains unchanged at 0 m. Since the water level in the vacuum breaker valve is variable according to the vacuum breaker valve model, the relative pressure of the gas in the valve after the vacuum breaker valve is opened is approximately maintained at atmospheric pressure.

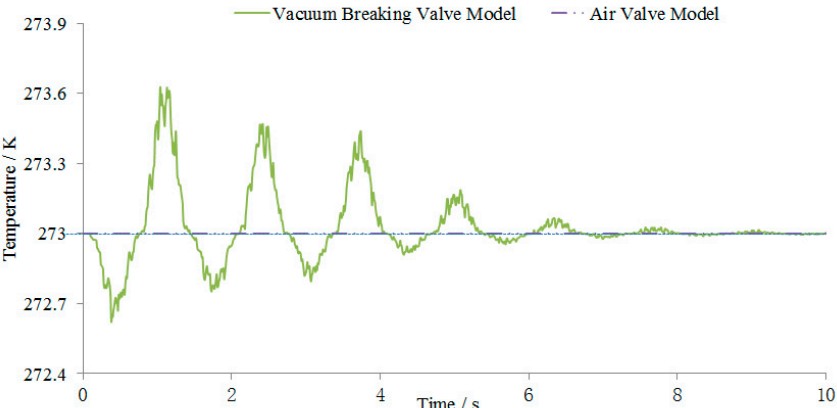

**Figure 7.** Temperature changes in the air in the valve.

According to Equations (1)–(4), different processes of the air pressure lead to different processes of the intake velocity change, as shown in Figure 6. When the air valve model is used to simulate the intake velocity of the vacuum breaker valve, the intake velocity gradually increases from 0, reaching a maximum value of 136.9 m/s at $t$ = 3.6 s, and then gradually decreases. At $t$ = 26.86 s, the intake velocity is less than 0, and the vacuum breaker valve begins to vent. As the water level is unchangeable, the calculated water level is always greater than the actual water level during the intake process; therefore, the gas pressure in the valve is too small, and the intake air velocity is continuously increased to a larger number before slowly being reduced until gas is discharged. The intake and exhaust period of the vacuum breaker valve is long. When the vacuum break valve model is used to simulate the intake velocity of the valve, the gas periodically enters and exits the vacuum break valve with the water hammer pressure fluctuation. The maximum intake and exhaust rates are small; the maximum intake rate does not exceed 22.5 m/s, the minimum value is not lower than −22.5 m/s, and the period is short.

According to Equation (28), different air pressure change processes lead to different temperature change processes, as shown in Figure 7. When the air valve model is used to numerically simulate the gas temperature in the vacuum breaker valve, the thermodynamic change in the gas in the valve obeys the ideal gas isothermal process, and the ambient temperature is maintained regardless of the temperature change of the valve gas. Thus, the valve gas temperature is kept constant at 273 K. When the vacuum breaker valve model is used for simulation, the gas temperature in the valve changes periodically with the gas pressure. The maximum temperature is 273.63 K and the minimum is 272.61 K, which is a small change. For this project, since the gas pressure fluctuation is not large, the change in the calculated temperature is also small, and thus, the influence of temperature is negligible. The above analysis shows that the calculation results are reasonable and consistent with common sense.

## 4. Discussion

The key point of this study is to derive a vacuum breaker valve model with a large intake air condition and apply it to actual water supply projects. In many previous studies, the air valve

model was used to numerically simulate the vacuum breaker valve in actual engineering. When the vacuum breaker valve is applied to the negative pressure protection of the general pipe or container, the assumptions of the air valve model are satisfied, due to the small intake and exhaust air condition. In fact, when the amount of intake air is large, it is difficult to fully realize heat exchange between gas and water during the flow of gas into and out of the vacuum destruction valve, so it is unreasonable to follow the ideal gas isothermal process in the thermodynamic change in the gas in the air valve. At this time, the air valve model cannot be used to simulate a vacuum breaker valve with a large intake air amount. For the air valve model, since the water level is assumed to be constant, the calculated water level is always higher than the actual water level during the intake process. As a result, the calculated gas pressure is too small, so there is a second gas pressure drop and recovery process. The result arises because the air valve model is assumed to be inconsistent with more intake conditions, and it is not a true gas pressure change process in the valve. The calculation results are likely to cause an engineer to misjudge that the size of the designed vacuum breaker valve is too small and replace it with a larger one, causing waste. For a vacuum breaker valve on the siphon outlet pipe, the air valve model cannot be applied due to the large intake air condition. This study adjusts the assumptions and derives the vacuum breaker valve model, making it suitable for more intake conditions based on the air valve model. It is assumed that the gas temperature in the valve is no longer atmospheric temperature but that it follows the isentropic process, and the change in water level is considered. The proposed vacuum breaker valve model (28) shows that the air valve model can be regarded as a special case under the condition that the vacuum breaker valve model has less intake air, and Equation (28) can be simplified to the same form as Equation (5). It can be seen from the calculation results that the vacuum breaker valve model is closer to the actual situation of the gas pressure, intake rate and temperature in the valve.

## 5. Conclusions

The vacuum breaker valve model in large air mass conditions is proposed in this paper based on the air valve model according to the modified assumptions. It is proved that there is no contradiction among these modified assumptions. In this case study, the vacuum breaker valve is simulated and analyzed by both the air valve model and the vacuum breaker valve model respectively. The results indicated that if the vacuum breaker valve model is used for calculation, the relative pressure of the gas in the valve is almost constant at atmospheric pressure, and the gas temperature and intake velocity periodically change with minor changes. However, if the air valve model is used, the relative pressure of the gas in the valve will decrease and then gradually rise to 0 m. The gas temperature remains constant at an ambient temperature of 273K. The intake velocity continues to increase to a larger value and then slowly decreases to 0 m/s. According to the discussion, the differences between the relative pressures, the gas temperatures and intake velocities are the results of different assumptions on the change in the thermodynamic and the water level. The air valve model is not applicable as the water level is constant, and higher than the actual water level during the intake process. The calculated gas pressure then is too small, causing the size of the vacuum breaker valve to be incorrectly judged. Therefore, compared with the air valve model, the vacuum breaker valve model is more accurate and suitable in large air mass conditions. Setting an effective and appropriate protective device is a significant research, because the security of the water supply engineering is very important. We all hope that in the future, we can make more progress on this work.

**Author Contributions:** Conceptualization, J.Z. and X.-y.Z.; methodology, X.-y.Z., C.-y.F., and X.-d.Y.; investigation, C.-y.F., and J.Z.; validation, X.-d.Y.; writing—original draft, X.-y.Z.; and C.-y.F.; writing—review and editing, X.-d.Y., T.-y.X., and J.-w.L.

**Funding:** This research was funded by the National Key R&D Program of China (Grant No. 2016YFC0401810), the National Natural Science Foundation of China (Grant No. 51879087) and the National Natural Science Foundation of China (Grant No. 51839008).

**Acknowledgments:** The authors would like to thank the National Natural Science Foundation of China (NSFC) for providing partial funding for the project.

**Conflicts of Interest:** The authors declare no conflict of interest. The funders had no role in the design of the study; in the collection, analyses, or interpretation of data; in the writing of the manuscript; or in the decision to publish the results.

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
