# Peer review of "Study on the Mathematical Model of Vacuum Breaker Valve for Large Air Mass Conditions"

_water, doi:10.3390/w11071358_

Round 1

Reviewer 1 Report

All the comments are provided in the attached file. 

Author Response

We would like to thank you for giving us a chance to resubmit our paper, and also truly grateful to the reviewers for giving us constructive suggestions. These suggestions indeed help us greatly to improve the scientific quality of the paper. Here we submit a new version of our manuscript with the title “Study on the Mathematical Model of Vacuum Breaker Valve for Large Air Mass Conditions”, which has been carefully modified on the original manuscript according to the reviewers’ suggestions, both in the content of the paper and the language as well. Our manuscript ID is water-519546

Reviewer 2 Report

General comment.

The authors present a mathematical model for the representation of Vacuum breaking Valves (VBV), in contrast to the classic models of air valve (AV) representation. In the article they try to show the different mode of operation of the VBV as opposed to the AV. However, in my opinion, this intention shown by the authors in the title and in the abstract is not reflected in the manuscript.

In general, the study of the background is quite deficient. The explanation of the background of the work is extremely brief and there are hardly any references to previous experiences in the literature. At the same time, from the reading of the introduction of the manuscript the objective of the work is not clear.

In general, my main concerns about the work are:

- From my point of view there is no conceptual difference in the operation of a VBV and an AV when air is admitted. That is why it is not clear in the work what the authors try to represent.

- For AVs, the authors use the MOC and the classic modeling of the behavior of the AVs based on the isentropic flow. Alternatively, they propose a methodology for VBV. In my opinion, some of the expressions used by the authors are thermodynamically valid for closed systems, although the air intake flow in a pipeline is a clearly open system in which the air mass is changing.

- In both cases, the resolution form is a conventional MOC, which hardly implies any scientific contribution. The possible scientific contribution of the work is the different representation of the VBV, although this model is not clearly explained. From my point of view there is a misunderstanding of the authors, since they try to explain a different mode of behavior of the air in both mechanisms (VBV and AV). The reality is that conceptually they are devices that admit air and therefore the model of behavior of the air should be the same in both cases.

- The case study presented is a laboratory test whose results do not provide conclusions. It seems that the response times of the VA used are lower than those of the VBV. In order to compare the results of both devices, they should have the same capacity and response times.

- There are also some formal deficiencies in the manuscript. Specifically, figures 1 and 3 do not have sufficient quality. Also, paragraphs with different line spans appear.

In summary, the work does not meet the minimum conditions to be accepted for publication. It is recommended to make deep changes in the work, clarifying the differences between AV and VBV. Likewise, the article must have a scientific presentation of the state of the art according to what is current in this type of journals.

For all these reasons, I recommend to the authors a thorough remodeling of the submitted manuscript, either as major changes of the manuscript or as a new resubmission.

Author Response

We would like to thank you for giving us a chance to resubmit our paper, and also truly grateful to the reviewers for giving us constructive suggestions. These suggestions indeed help us greatly to improve the scientific quality of the paper. Here we submit a new version of our manuscript with the title “Study on the Mathematical Model of Vacuum Breaker Valve for Large Air Mass Conditions”, which has been carefully modified on the original manuscript according to the reviewers’ suggestions, both in the content of the paper and the language as well. Our manuscript ID is water-519546.

Round 2

Reviewer 1 Report

The paper has been considerably improved by mentioning more similar studies and results. Some writing improvements are required, for example, line 66 "... as you need."  is not common scientific writing. Most of the new parts must be considered for the correction of English writing. I still believe that the final conclusions are not sound and rigid and mostly provides common facts. However, this paper can be interesting for the readers by provides the formulations clearly and highlighting the inherent differences between air valves and vacuum breaker valves. 

Author Response

We would like to thank you for giving us a chance to resubmit our paper, and also truly grateful to the reviewers for giving us constructive suggestions. These suggestions indeed help us to improve the scientific quality of the paper. Here we submit a new version of our manuscript with the title “Study on the Mathematical Model of Vacuum Breaker Valve for Large Air Mass Conditions”, which has been carefully modified on the original manuscript according to the reviewers’ suggestions, both in the content and the language of the paper. Our manuscript ID is water-519546.

The following is a point-by-point response to the comments/suggestions of the reviewers.

 Thanks for your comments. We are sorry for the mistakes in English writing, and we have asked native English speakers to correct our English writing. We have improved the English writing and modified both the introduction part and the conclusion part in the revised paper .

Reviewer 2 Report

No Comment

Author Response

Comments and Suggestions for Authors
No Comment

Thank you very much for your recognition of this article.